# Stress-Induced Microcracking and Fracture Characterization for Ultra-High-Temperature Ceramic Matrix Composites at High Temperatures

**DOI:** 10.3390/ma15207074

**Published:** 2022-10-11

**Authors:** Mingyu Gu, Chunyan Wu, Xingyu Chen, Yu Wan, Yumeng Liu, Shan Zhou, Hongwei Cai, Bi Jia, Ruzhuan Wang, Weiguo Li

**Affiliations:** 1Chongqing Key Laboratory of Nano-Micro Composite Materials and Devices, School of Metallurgy and Materials Engineering, Chongqing University of Science and Technology, Chongqing 401331, China; 2College of Aerospace Engineering, Chongqing University, Chongqing 400030, China

**Keywords:** ultra-high-temperature ceramic matrix composites, stress-induced microcracking, fracture strength, temperature, theoretical model

## Abstract

In this paper, we estimated the temperature-dependent critical inclusion size for microcracking under residual stress and applied stress for particulate-reinforced ultra-high-temperature ceramic matrix composites. The critical flaw size and applied stress for the stable growth of radial cracks under different temperatures were also estimated. It was found that under a lower applied stress, the critical inclusion size was sensitive to the temperature. Under higher applied stresses, the sensitivity became smaller. For ceramic materials with pre-existing microcracks, the crack resistance could be improved by increasing the service stress when the service stress was low. As the temperature increased, the critical flaw size of the materials decreased; the applied stress first increased and then decreased. Finally, a temperature-dependent fracture strength model of composites with a pre-existing critical flaw was proposed. A good agreement was obtained between the model prediction and the experimental data. In this work, we show a method for the characterization of the effects of temperature on the fracture behavior of ceramic-based composites.

## 1. Introduction

Ultra-high-temperature ceramics (UHTCs) such as transition metal borides and carbides have melting points higher than 3000 °C; these can be used in high-temperature and oxidizing environments and have a good chemical and physical stability [1,2,3,4]. The addition of a reinforcing phase has been proven to be the most promising way of improving the mechanical properties of UHTCs [3,4,5,6,7,8]. Particulate-reinforced UHTC matrix composites have demonstrated a good combination of fracture strength, fracture toughness, hardness and oxidation resistance. Composites have been developed for leading edge and nose cap materials in hypersonic vehicles.

When the thermal expansion coefficient of a particle is different from that of a matrix, a residual stress field is developed within and around the particles as the materials cool from the sintering temperature. When the thermal expansion coefficient of a particle is smaller than that of a matrix, tangential tensile stress outside the particle is formed and can initiate radial cracks as the inclusion size reaches a critical value. A radial crack is very deleterious to the strength of composites as it is more likely to link with other grains and become a major flaw [9,10]. A radial crack would be a part of the critical flaw, causing the catastrophic failure of the composites. Green estimated the critical inclusion size for microcracking at second phase inclusions due to applied stress and residual stress by using fracture mechanics [10]. The work of Green reported that when the value of applied stress was similar to residual stress, the critical inclusion size could be greatly reduced and then a microcrack arrest would become less likely. By estimating the stress intensity factor at the tip of an annular flaw associated with a spherical particle subjected to residual stress, Krstic and Vlajic estimated the critical inclusion size for spontaneous microcracking [11]. They considered the effects of residual stresses acting along the grain boundary and the flaw face. The results showed that materials containing long cracks possessed a higher resistance to internal cracking.

However, few researchers have reported the effects of temperature on the critical inclusion size for stress-induced microcracking at second phase inclusions due to applied stress and residual stress. As the temperature increases, the residual stress reduces and the material performances and microstructures will also change. These factors affect the critical size. Considering the high-temperature applications of particulate-reinforced UHTC matrix composites, it is important and necessary to study the control mechanisms of microcracking and the fracture of composites at high temperatures. At present, the theoretical work mainly focuses on the high-temperature fracture strength of UHTC matrix composites; theoretical methods that can effectively characterize the effect of temperature and critical flaw size on the fracture strength of materials have been proposed [12,13,14,15,16]. However, few researchers have established a theoretical relationship between the critical inclusion size for stress-induced microcracking and temperature.

In this work, we estimated the critical inclusion sizes for microcracking at second phase inclusions due to applied stress and residual stress at high temperatures based on classical fracture theories and a temperature-dependent fracture toughness model. We also estimated the upper ranges of flaw size and applied stress for stable crack growth under different temperatures. Furthermore, a temperature-dependent fracture strength model of composites with pre-existing critical flaws was developed. A good agreement was obtained between the model prediction and the experimental data.

## 2. Temperature-Dependent Models

For particulate-reinforced ceramic matrix composites, when the thermal expansion coefficient of the particle is smaller than that of the matrix, tangential tensile residual stress outside the particle is formed on cooling from the sintering temperature and may initiate radial cracks. In this work, the particle was set to be a spherical particle. The solid model was the annular flaw emanating from the spherical particle interface (as shown in Figure 1). All the models developed were based on the conditions of quasi-static loading at a constant temperature.

The residual stress could be expressed as follows [17]:(1)$$P=\left(\alpha_{m}-\alpha_{p}\right)\left(T_{s}-T_{r}\right){\left(\frac{1+\nu_{m}}{2E_{m}}+\frac{1-2\nu_{p}}{E_{p}}\right)}^{-1}$$
where *α*_m_ and *α*_p_ are the thermal expansion coefficients of the matrix and particle, respectively; *E*_m_ and *E*_p_ are the Young’s modulus of the matrix and particle, respectively; *ν*_m_ and *ν*_p_ are the Poisson’s ratio of the matrix and particle, respectively; *T*_s_ is the sintering temperature; and *T*_r_ is the room temperature. The radial and tangential components of the residual stress could be expressed as follows:(2)$$\sigma_{r}^{P}=-P{\left(\frac{R}{r}\right)}^{3}=-\left(\alpha_{m}-\alpha_{p}\right)\left(T_{s}-T_{r}\right){\left(\frac{1+\nu_{m}}{2E_{m}}+\frac{1-2\nu_{p}}{E_{p}}\right)}^{-1}{\left(\frac{R}{r}\right)}^{3}$$
(3)$$\sigma_{\theta }^{P}=\frac{P}{2}{\left(\frac{R}{r}\right)}^{3}=\frac{1}{2}\left(\alpha_{m}-\alpha_{p}\right)\left(T_{s}-T_{r}\right){\left(\frac{1+\nu_{m}}{2E_{m}}+\frac{1-2\nu_{p}}{E_{p}}\right)}^{-1}{\left(\frac{R}{r}\right)}^{3}.$$

The study showed that the stress intensity factor at the tip of the flaw was related to the residual stresses acting along the grain boundary and the flaw face [11]. By superimposing these stresses, the stress intensity factor at the tip of the flaw due to residual stresses could be obtained as follows [11]:(4)KIT=2PR12π121+s/R121−1−11+s/R212+121+s/R321−11+s/R212.

As the temperature increased, the residual stress was relieved and the material properties changed; therefore, Equation (1) could be given as follows [12,13,14]:(5)$$P\left(T\right)=\left(\alpha_{m}\left(T\right)-\alpha_{p}\left(T\right)\right)\left(T_{s}-T\right){\left(\frac{1+\nu_{m}\left(T\right)}{2E_{m}\left(T\right)}+\frac{1-2\nu_{p}\left(T\right)}{E_{p}\left(T\right)}\right)}^{-1}$$
where *T* is the current temperature; *α*_m_(*T*) and *α*_p_(*T*) are the temperature-dependent thermal expansion coefficients of the matrix and particle, respectively; *E*_m_(*T*) and *E*_p_(*T*) are the temperature-dependent Young’s modulus of the matrix and particle, respectively; and *ν*_m_(*T*) and *ν*_p_(*T*) are the temperature-dependent Poisson’s ratio of the matrix and particle, respectively.

Submitting Equation (5) into Equation (4), the temperature-dependent stress intensity factor at the tip of the flaw due to residual stress could be obtained as follows:(6)KITT=2PTR12π121+s/R121−1−11+s/R212+121+s/R321−11+s/R212.

According to Equation (6), the temperature-dependent critical inclusion size for spontaneous microcracking could be obtained as follows:(7)RCT=πKC′T24PT21+α121−1−11+α212+121+α321−11+α212−2
where α = *s*/*R* and $K_{C}^{\prime }\left(T\right)$ is the temperature-dependent critical stress intensity factor for a local failure. Equation (7) shows the critical size of a grain below which a microfracture could not occur. $K_{C}^{\prime }\left(T\right)$ was equal to or lower than the temperature-dependent critical stress intensity factor for an extension of critical flaw $K_{C}^{}\left(T\right)$. Green used a parameter to express the ratio of these two properties under a low temperature [10]. However, at present, few studies have reported relative research for $K_{C}^{\prime }\left(T\right)$. It is difficult to determine the value of $K_{C}^{\prime }\left(T\right)$. In this work, and following Green’s method [10], a parameter β was introduced to express the ratio of $K_{C}^{\prime }\left(T\right)$ and $K_{C}^{}\left(T\right)$, which was thus equal to or lower than 1.

The temperature-dependent fracture toughness of composites could be obtained as follows [12]:(8)KIC=2γTET1−νT212
where *E*(*T*) is the temperature-dependent Young’s modulus of the composites and $\nu \left(T\right)$ is the temperature-dependent Poisson’s ratio of the composites. The two basic material parameters could be easily obtained by experiments or by the theory of single-phase inclusion.

Submitting Equation (8) into Equation (7), the temperature-dependent critical inclusion size for spontaneous microcracking could thus be expressed as follows:(9)RCT=πβKIC24PT21+α121−1−11+α212+121+α321−11+α212−2.

When the external stress $\sigma_{a}$ was applied to the solid model, the stress intensity factor at the tip of the flaw due to applied stress could be obtained as follows [10]:(10)KIS=2σasπα+2α+112.

The stress intensity factor was obtained by superimposing the prior stress distribution along the flaw face. By combining Equations (6) and (10), it was possible to estimate the temperature-dependent stress intensity factor at the tip of the flaw due to both residual stress and applied stress; this could be expressed as follows:(11)KT=2PTR12π121+s/R121−1−11+s/R212+121+s/R321−11+s/R212+2σasπα+2α+112

The applied stress acted to reduce the critical inclusion size for microcracking. Based on Equation (11), the temperature-dependent critical inclusion size for microcracking under the application of external stress and residual stress could be obtained as follows:(12)RCS=βKIC2π4PT1+α121−1−11+α212+121+α321−11+α212+σaαπα+2α+1122

It could be found that the critical inclusion size for microcracking was controlled by the ratio of the flaw size and the grain size.

After microcracking occurs, the radial crack can become unstable. However, if the applied stress is low, a crack arrest would occur. The crack would be stationary without an increase in applied stress. With an increase in applied stress, the crack would grow again. The crack may then undergo a stable growth until the crack size and applied stress reach the critical values of *s*_C_ and $\sigma_{C}$. At this point, the derivative of the stress intensity factor at the tip of the crack would equal 0. After then, the crack would become unstable again, leading to the failure of the materials. This is the point of failure of materials. The critical applied stress is, therefore, the fracture strength of the materials. The critical crack size and applied stress could be determined by the following functions:(13)$$\left\{\begin{array}{l} K\left(T\right)=K_{IC}\\ \frac{dK\left(T\right)}{ds}=0 \end{array}\right..$$

In general, critical flaws often form in the microstructure of prepared materials. Studies have shown that the strength of UHTC-based composites is controlled by added inclusion clusters [18,19]. The critical flaw size of UHTC-based composites for a failure is equal to the size of the largest inclusion cluster. In this work, the inclusion cluster was simplified to be a spherical inclusion with an annular flaw emanating from the inclusion; 2(*R* + *s*) was then equal to the critical flaw size. The problem converted to be the failure of the materials with pre-existing critical flaws. The failure occurred when this pre-existing flaw began to extend. Only if the total stress intensity factor at the tip of this pre-existing flaw reached the fracture toughness of the composites would the fracture of the composites occur. When KT=KIC=2γTET1−νT212,σa=σf,, a temperature-dependent fracture strength model of UHTC-based composites with pre-existing critical flaws could be obtained as follows:(14)σf=πγ0ETα+121−νT2sα+21−∫0TCpTdT∫0TmCpTdT+ΔHM12−AΦtPTα+1αα+212
(15)Φt=1+s/R121−1−11+s/R212+121+s/R321−11+s/R212
where *A* is a constant related to the stress relaxation [12].

## 3. Results and Discussion

In this work, we used ZrB_2_–SiC composites to study the critical inclusion size for stress-induced microcracking at high temperatures. Watts et al. reported that spontaneous microcracking occurred at room temperature when the size of the SiC particle embedded in the microstructure of ZrB_2_–SiC composites was larger than 11.5 µm [6]. Using Equation (9) and the critical inclusion size for spontaneous microcracking at room temperature, the value of *β* was calculated and was assumed to be a constant under different temperatures. In the calculations, the relative parameters were determined by experiments and are shown in Table 1 [12,20,21,22,23,24]. As the temperature had little effect on the Poisson’s ratio of the UHTCs, the Poisson’s ratio of the materials at room temperature was used in the calculation process [12,13,14,15,16]. The effects of small changes to the Poisson’s ratio with temperature on the temperature dependence of the mechanical properties and the fracture behavior of the materials were ignored.

Figure 2 shows the critical inclusion sizes for spontaneous microcracking under different temperatures for *s*/*R* = 0.1. We found that the critical inclusion size for spontaneous microcracking increased in general as the temperature increased. When the temperature was lower than about 500 °C, the critical inclusion size for spontaneous microcracking significantly increased. When the temperature was in the range of about 500 °C to 900 °C, the critical inclusion size for spontaneous microcracking decreased. Yet, the degree of degradation was very small and the critical inclusion size was also higher than that at a low temperature. As the temperature increased above about 1000 °C, the critical inclusion size for spontaneous microcracking remarkably increased. The above changing trend of the critical inclusion size for spontaneous microcracking with temperature was controlled by the combined effects of the temperature on the residual stress and fracture toughness. It was observed that the effect of residual stress on the material strength became very small above 1000 °C. This coincided with the experimental report that ZrB_2_–20%SiC composites maintained a great fracture strength around 1300 °C due to residual stress recovering [25].

Figure 3 shows the predicted critical inclusion sizes for stress-induced microcracking under different temperatures and applied stresses for *s*/*R* = 0.1. It shows that when $\sigma_{a}$ = 0.2 and 0.4*P* (*T* = 25 °C), the critical inclusion sizes for stress-induced microcracking first increased quickly and then decreased a little as the temperature increased, followed by a remarkable increase. At ultra-high temperatures (above about 2000 °C), the critical inclusion size decreased. For the materials without microcracks owing to residual stress, microcracking could occur under a low applied stress; yet, with an increase in temperature, crack arrest would occur. It was observed that under a low applied stress, the critical inclusion size for stress-induced microcracking was very sensitive to the temperature. As the applied stress increased, the sensitivity decreased. When the applied stress was of the order of the residual stress, the temperature had little effect on the critical inclusion size for stress-induced microcracking. This suggested that a higher applied stress would cause extensive microcracking under different temperatures. It was found that under a low applied stress, the relationship between the critical inclusion size for microcracking and the temperature was mainly controlled by the relaxation of residual stress; under a higher applied stress, it was mainly affected by the applied stress and the local fracture toughness. The results showed that for ceramic materials, the presence of microcracks must be suppressed and the effects of service stress must be taken into account in the microstructure design. When service stress is low, increasing the service temperature could improve the crack resistance.

Figure 4 and Figure 5 show the critical points for instable crack growth after the initiation and stable propagation of a radial crack under different temperatures for *R* = 10 μm. Figure 4 shows that the critical flaw size for stable crack growth decreased as the temperature increased. This suggested that the region of stable crack growth decreased with an increase in temperature. As shown in Figure 5, the critical applied stress for stable crack growth first increased and then decreased as the temperature increased. This critical applied stress could be assumed to be the fracture strength of the materials, the failure of which was caused by the initiation and propagation of the radial crack. It was observed that the first increase in this strength with temperature was owing to the relaxation of residual stress and the decrease in the critical flaw size. With a further increase in temperature, the negative effects of the temperature led to the degradation of the material strength again. As residual stress is greatly released at high temperatures [6,19], then it has very little effect on the fracture of materials. Thus, at high temperatures the microcracking requires a much higher applied stress compared with residual stress. Under this condition, stable crack growth would not occur. Figure 4 and Figure 5 also indicate that if a stable crack growth region existed, the fracture strength of the materials at a high temperature would be higher than that under a low temperature owing to the relaxation of residual stress. 

Figure 6 shows the comparison between the predicted temperature-dependent fracture strength by Equation (14) and the measured data of ZrB_2_–30vol%SiC–2vol%B_4_C composites with pre-existing critical flaws. In the work of [19], the temperature-dependent Young’s modulus and critical flaw size of the composites were also measured. Thus, during the calculation, the measured Young’s modulus and critical flaw size at a high temperature were used [12,19]. The used inclusion (particle) size of the composites was 1.9 µm, according to the measured data [19]. Figure 6 shows that a good agreement was obtained between the model prediction and the experimental data. It could be indicated that for materials with a pre-existing critical flaw, a catastrophic failure at a high temperature occurs when the flaw propagates. Therefore, a reduction in the initial flaws should be first considered when designing materials.

## 4. Conclusions

In this work, we developed effective theoretical models for depicting the effects of temperature, critical flaw size and residual stress on the fracture behavior of UHTC matrix composites, including spontaneous microcracking, stress-induced microcracking, stable crack growth and fracture strength. When the material parameters such as the temperature-dependent Young’s modulus, critical flaw size, melting point and thermal expansion coefficient of the composites were known, the models could be easily used to predict the fracture properties of the composites at high temperatures. The predictions showed that the temperature-dependent critical inclusion size for microcracking was mainly controlled by the values of the temperature, pre-existing flaw size, applied stress and residual stress. A reduction in the initial flaws should be first considered when designing materials, which would result in higher strength values at high temperatures. An excellent agreement was obtained between the model prediction and the experimental data. This indicated the rationality and applicability of the proposed method, considering the effects of temperature on the fracture behavior of composites.

## Figures and Tables

**Figure 1 materials-15-07074-f001:**
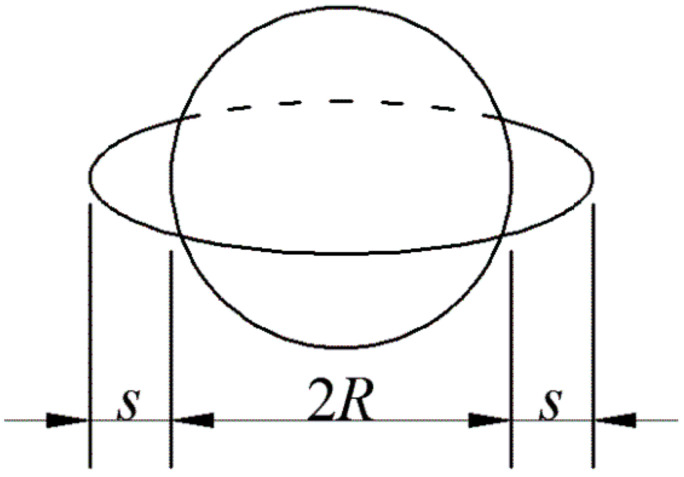
An annular flaw emanating from the spherical particle (*R* is the particle radius and *s* is the flaw size).

**Figure 2 materials-15-07074-f002:**
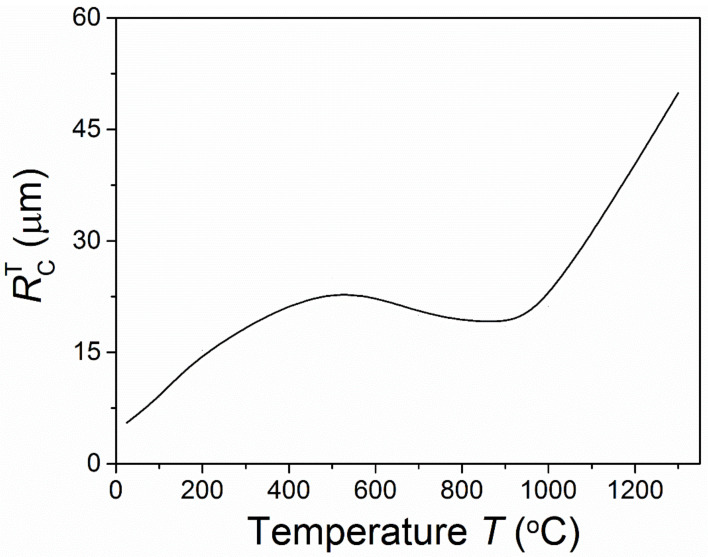
Temperature-dependent critical inclusion size for spontaneous microcracking.

**Figure 3 materials-15-07074-f003:**
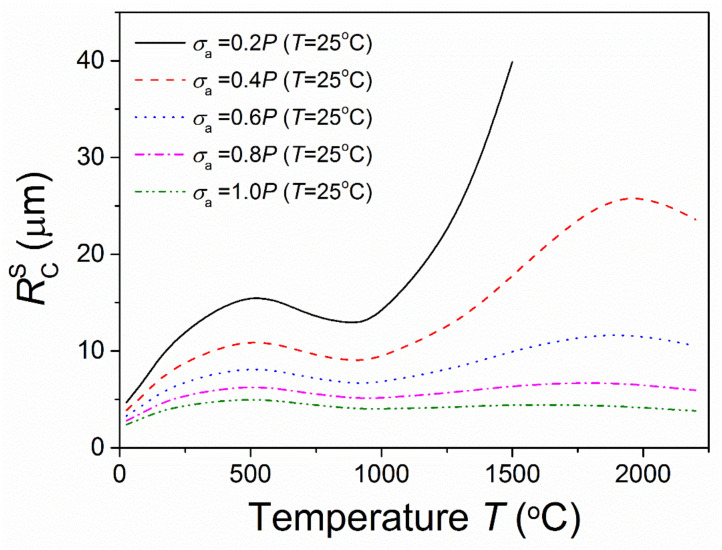
Temperature-dependent critical inclusion size for stress-induced microcracking corresponding with different applied stresses.

**Figure 4 materials-15-07074-f004:**
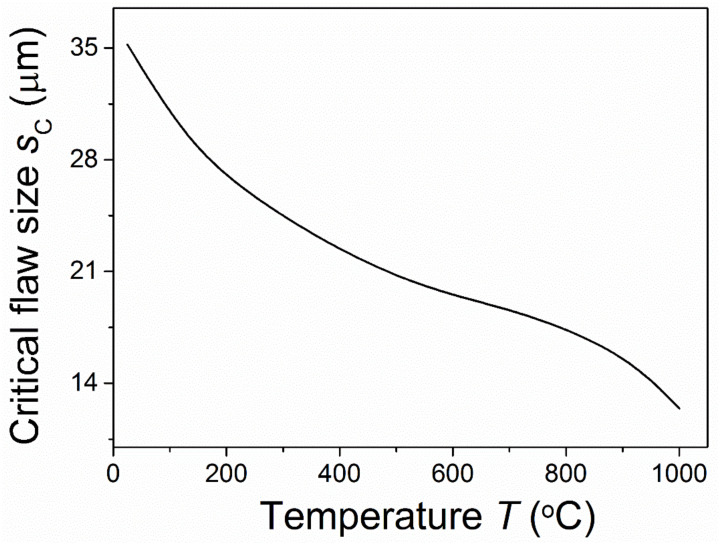
Temperature-dependent critical flaw size for stable crack growth.

**Figure 5 materials-15-07074-f005:**
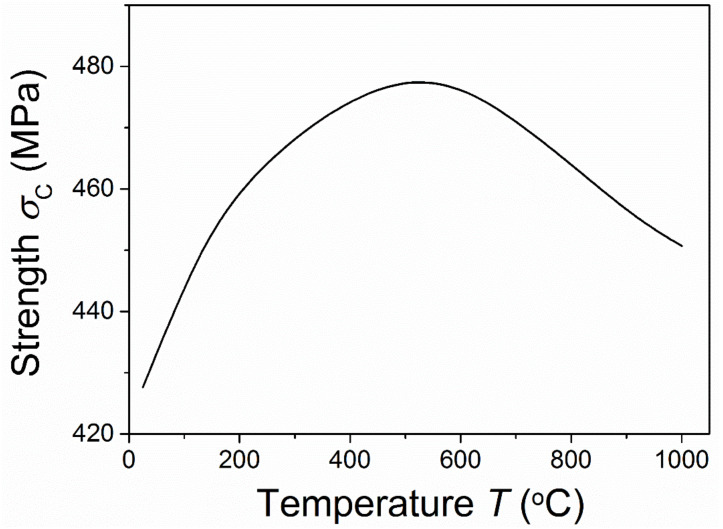
Temperature-dependent critical applied stress (strength) for stable crack growth.

**Figure 6 materials-15-07074-f006:**
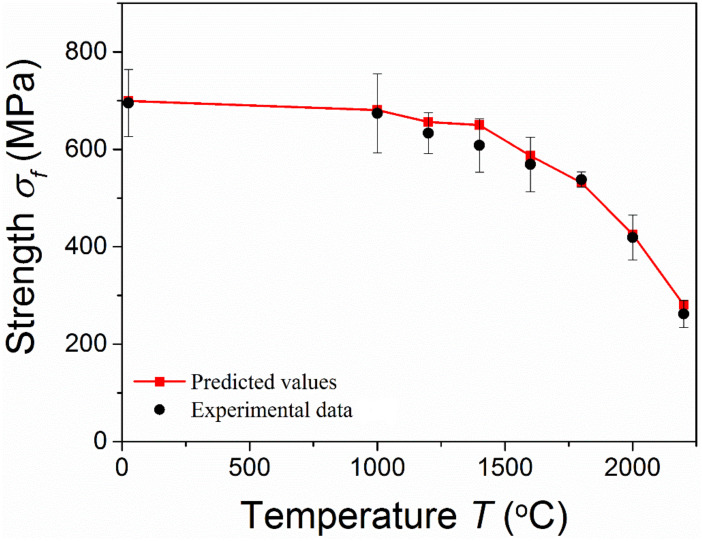
Temperature-dependent fracture strength of ZrB_2_–30%SiC–2vol%B_4_C [19].

**Table 1 materials-15-07074-t001:** Material properties [12,20,21,22,23,24].

Material Parameters	Values and Expressions
ν_m_	0.25
ν_p_	0.19
*T*_s_ (°C)	1950
*T*_m_ (°C)	3049.85
*T*_0_ (°C)	25
*γ*(*T*_0_) (J⋅m^−2^)	12.76
*C*_p_(*T*) (cal/mol)	15.34 + 2.25 × 10^−3^ (*T* + 273.15) − 3.96 × 10^5^ (*T* + 273.15)^−2^
*αm*(*T*) (°C^−1^)	(2.33 + 0.006 × (*T* + 273.15) − 0.2 × 10^−5^ × (*T* + 273.15)^2^) × 10^−6^
*α*_p_(*T*) (°C^−1^)	(−1.8276 + 0.0178(*T* + 273.15) − 1.5544 × 10^−5^(*T* + 273.15)^2^ + 4.5246 × 10^−9^ (*T* + 273.15)^3^) × 10^−6^(0 °C ≤ *T <* 1000 °C)
*α*_p_(*T*) (°C^−1^)	5.0 × 10^−6^(*T* ≥ 1000 °C)
*E*_m_(*T*) (GPa)	$507.0-2.54T\exp\left(-\frac{T_{m}}{T}\right)+1.9\left(T-0.363T_{m}+\left|T-0.363T_{m}\right|\right)\exp\left(-\frac{T_{m}}{T}\right)$
*E*_p_(*T*) (GPa)	$460.0-0.04\left(T+273.15\right)\exp\left(-\frac{962}{T+273.15}\right)$

## Data Availability

Data is contained within the article.

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
