# Peer review of "Stress-Induced Microcracking and Fracture Characterization for Ultra-High-Temperature Ceramic Matrix Composites at High Temperatures"

_materials, 2022, doi:10.3390/ma15207074_

Round 1
Reviewer 1 Report
Stress-induced microcracking and fracture characterization for ultra-high ceramic matrix composites at high temperatures – M. Gu et al.
General Comments: In this manuscript, the stress intensity factor is obtained by including the temperature-dependent material card. Mainly, the coefficient of thermal expansion, Young’s modulus, and Specific heat have been incorporated into the closed-form expressions. This is a unique approach to computing the critical flaw size and thereby determining the critical stress that is dependent on the temperature. My specific comments are provided below.
Specific Comments:
1. Please revise the entire manuscript to rectify grammatical errors.
2. The Poisson’s ratio has been taken as temperature-dependent. Yet, this has not been incorporated in the Results and Discussion section. The material properties in Table 1 assume a constant value. Please explain.
3. Line 174: the comment on Poisson’s ratio variation w.r.t temperature cannot be generalized to the larger materials community. Although this change might not be drastic when compared to E-modulus, it is encouraged to present a bound-based analysis [± 1%] or modify the statement.
4. Can the model capture stress with incremental temperature changes? Or is this intrinsically a quasi-static model with load applied at a constant temperature? Please include this in the discussion section as this is not obvious.
5. Line 231: Please refer to the figure numbers.
Reviewer 2 Report
The authors have conducted interesting research on fracture mechanics of high temperature ceramic based composites. However, it should be revised in the light of the following observations:1. The authors mention that the effect of residual stress on material strength becomes very little above 1000oC. How does this relate to the melting temperature ZrB2 and SiC.2. The authors should provide temperature dependent mechanical properties from published literature and correlate with inclusions size and mechanical properties.3. There are several language usages errors that require proofreading by an English speaker/professor. Also, the language is quite informal. The authors should adhere to a formal technical reporting format and first-person references including I, we, he, she, they etc. should be avoided.4. A list of nomenclature should be included before the introduction section.5. A table of mechanical properties of prospective high temperature ceramic based composites and their constituent materials should be provided.6. Conclusions section should be revisited and strengthened to highlight the novelty of this researchThe authors have conducted interesting research on fracture mechanics of high temperature ceramic based composites. However, it should be revised in the light of the following observations:
1. The authors mention that the effect of residual stress on material strength becomes very little above 1000oC. How does this relate to the melting temperature ZrB2 and SiC.2. The authors should provide temperature dependent mechanical properties from published literature and correlate with inclusions size and mechanical properties.3. There are several language usages errors that require proofreading by an English speaker/professor. Also, the language is quite informal. The authors should adhere to a formal technical reporting format and first-person references including I, we, he, she, they etc. should be avoided.4. A list of nomenclature should be included before the introduction section.5. A table of mechanical properties of prospective high temperature ceramic based composites and their constituent materials should be provided.6. Conclusions section should be revisited and strengthened to highlight the novelty of this research
